# The Importance of Optimal Gaze Direction on Deep Neck Flexor Activation in Chronic Neck Pain

**DOI:** 10.3390/healthcare8040449

**Published:** 2020-11-01

**Authors:** Byoung-Kwon Lee, Dong-Kwon Seo

**Affiliations:** Department of Physical Therapy, Konyang University, Daejeon 35365, Korea; lbk6326@konyang.ac.kr

**Keywords:** deep neck flexor, sternocleidomastoid, cranio-cervical flexion test, gaze direction, chronic neck pain

## Abstract

Chronic neck pain (CNP) patients have weak deep neck flexors (DNF) and a hyperactive sternocleidomastoid (SCM). The cranio-cervical flexion test (CCFT) promotes activation of the DNF and decreases activity of the SCM, promoting pain recovery, but research suggests SCM activation increases with increasing gaze direction. We aimed to investigate how DNF and SCM activation varies according to gaze direction in the CCFT, and to prescribe the appropriate gaze direction for CNP. Twenty-eight CNP subjects had their maximum strength pressure level determined by CCFT for strength (20–~30 mmHg) and at each of the measured pressures, DNF and SCM thickness in each of four gaze directions (0°, 20°, 40°, and 60°) was measured by ultrasound imaging. The DNF to SCM ratio varied significantly according to gaze direction (*p* < 0.05), with gaze directions of 20° and 0° being significantly different from 40° (*p* < 0.05). Although there was no significant difference in DNF activation according to gaze direction, there was in SCM activation (*p* < 0.05), with SCM 60° significantly different from SCM 20° and SCM 40° (*p* < 0.05). In order to increase DNF activation efficiency during the CCFT, SCM activation should be controlled, and a gaze direction below 20° is the most efficient. This can inform DNF training of CNP patients in a clinical environment.

## 1. Introduction

Neck pain is a common complaint in modern humans and has highly variable recovery and recurrence rates. In about one-third of people, it develops into chronic neck pain (CNP) [1,2], and patients with neck pain suffer from dysfunction [3], headaches [4,5], and a reduced quality of life [1]. CNP patients display altered patterns of muscle activation, such as joint instability caused by capsular ligament laxity [6], and delayed onset of postural muscles [7,8,9]. CNP patients exhibit a weakness of the deep neck flexors (DNF)—the longus capitis and longus colli—and hyperactivity of superficial muscles—the sternocleidomastoid (SCM) and the anterior scalene [8,10,11]. Furthermore, there is increased tension in the suboccipital muscles [12,13], and increased muscle fatigue due to weakened DNF [14,15]. Many previous studies have demonstrated that CNP requires functional recovery of the DNF and reduced SCM activity in order for pain and disability to be reduced and postural control improved [16,17,18].

The cranio-cervical flexion test (CCFT) is a specific therapeutic exercise that restores functional roles such as spinal segmental stabilization [19]. The CCFT promotes activation of the DNF and decreases activity of the SCM, which is an effective method for achieving throat function and pain recovery [20,21,22,23,24,25,26]. The CCFT is a common clinical setting in which to perform cranio-cervical flexion without excessive contraction of the SCM [27,28]. In most previous studies, changes in CCFT muscle activation have been confirmed by electromyography, with CNP patients showing decreased activation of the DNF and increased activation of the SCM [8,19,22]. Recently, the negative correlation between DNF and SCM activity was confirmed through ultrasound imaging [26,29]. During CCFT performance, the gaze direction should be slightly downward [19]. A recent study on subjects without CNP reported that as the line of sight increased above the horizontal line (0°) to 20° and 40° during cranio-cervical flexion, both DNF and SCM activity increased positively [30]. This is different from previous studies showing that DNF and SCM activity are negatively related during CCFT performance, and instead, the activity of both muscles was shown to increase according to the gaze. In addition, this recent study suggested that SCM activation is affected by gaze direction during neck rotation [31].

The CCFT is an exercise method that aims to improve the function of the DNF in CNP patients, and it is important to accurately identify changes in DNF and SCM activity according to gaze direction, so that the appropriate gaze direction can be prescribed. Therefore, we hypothesized that DNF and SCM activity would gradually increase according to gaze direction, and that there would be a significant difference in the amount of change in their activity. The purpose of this study was to identify the optimal gaze direction in the CCFT for enhancing the DNF in CNP patients, so that CNP patients can recover normal neck function.

## 2. Materials and Methods

### 2.1. Participants and Procedure

This study was conducted between September and December 2019 in Daejeon City, South Korea. The type of study is cross-sectional design (Figure 1). Twenty-nine CNP patients were enrolled. Assuming an effect size of 0.25, a power level of 0.80, and a significance level (α) of 0.05, we used the general power program to calculate that a minimum number of 24 subjects were required. The inclusion criterion was to have experienced idiopathic pain between the suboccipital region and the T1 level, with a duration of at least 6 months and a frequency of no less than once a week [32]. The criteria for exclusion of subjects achieving less than 26 mmHg in the CCFT test [19] were: radicular (dermatomal pattern) pain with symptoms suggesting nerve root involvement when screened by the Spurling test [33,34], and clinical cervical instability (level 6) as assessed by a joint mobility test [35]. This study was conducted in accordance with the Declaration of Helsinki and received approval from the institutional review board of Konyang University (KYU-2019–331–01) and the Clinical Research Information Service (Clinical IRB KCT0005261). Written consent was obtained from all participants. This study recruited 31 subjects with neck pain through the recruitment announcement in the hospital lobby, outpatient department, and ward. Two subjects with clinical cervical instability (level 6) were excluded, and finally 29 subjects participated in the study. All subjects first had their maximum strength pressure level determined through the CCFT for strength test. At each measured pressure, the thickness of the DNF and SCM in each of the four gaze directions (0°, 20°, 40°, and 60°) was measured by ultrasound imaging, and the results were analyzed.

### 2.2. Outcome Measurements

#### 2.2.1. Joint Mobility Test

For the joint mobility test (JMT), a manual test (assessing both flexion/extension and lateral flexion) was conducted based on the rating system for joint mobility suggested by Olson (2001) [35,36]. Three senior physiotherapists, each with more than 20 years of experience, conducted the JMT. The flexion/extension test was performed with the subject in the side-lying position, and the lateral flexion test was performed with the subject in supine. The JMT evaluated the quantity and end-feel of movements by repeated flexion and extension, and repeated lateral flexion, from C7 to the first intersegment of the thoracic spine, and from C1 to the second intersegment. Previous research has shown inter-examiner reliability of between 70 and 87% and kappa coefficients ranging between 0.28 and 0.43, which is considered to be only ‘fair to moderate’ [36]. 

#### 2.2.2. CCFT for Strength 

The CCFT for strength test measures the strength of the DNF. With the subject in supine, an air-filled pressure sensor unit (Stabilizer™, Chattanooga Group Inc., Tennessee, USA) was placed between the subject’s neck and the bed, and the initial pressure was set to 20 mmHg. The subject was asked to maintain pressure at 20 mmHg by bending the neck, which was achieved by cranio-cervical flexion. At that time, the subject’s gaze should be directed downward, with the mouth slightly open [37]. The test consisted of 5 steps from 20 mmHg to 30 mmHg, and if the subject maintained the position for 10 s at each step, they progressed on to the next step. If the subject was not able to withstand 10 s at a given stage, the previous stage was identified as the subject’s maximum DNF strength [19,28]. Intraclass correlation coefficients ranged between 0.81 (activation score) and 0.93 (performance index) [19]. Cohen’s Kappa coefficient for the pooled data of all muscles was determined to be 0.61 [38].

#### 2.2.3. CCFT for Endurance

In the CCFT for endurance test, with the subjects in the same posture as for the CCFT for strength test, they raised their head 2.5 cm from the ground and the overlapped skin was marked in a straight line. In order to fit 2.5 cm under the back of the subject’s head, the crossing of the second and third fingers of the measurer were inserted and measured after training to make only a slight contact. The test ended when either the marked skin gaps or the position of the held head changed. After training, two measurements were taken, with a 5 min break between training and measurements. The measured time was recorded [39,40]. Intra-examiner reliability (intraclass correlation coefficient, ICC [1,3]) ranged between 0.82 and 0.91, and inter-examiner reliability (ICC [1,2]) ranged between 0.67 and 0.78 [41,42]. Cohen’s Kappa coefficient for the pooled data of all muscles was determined to be 0.61 [38].

#### 2.2.4. Neck Disability Index

The neck disability index (NDI) to measure the neck-pain intensity and disability included a 10-item questionnaire where each item is measured on a 6-point scale from 0 (no disability) to 5 (full disability). The final NDI score was classified into five levels: 0–4, no disability; 5–14, mild disability; 15–24, moderate disability; 25–34, severe disability; and ≥ 35, complete disability [43,44].

#### 2.2.5. Measurement of Muscle Thickness

Ultrasound imaging of neck flexor muscles (longus colli and the SCM) was conducted using a real-time ultrasound device (Sonoview, SAMSUNG, Korea, 2018) with an 8-MHz, 38.7-mm linear array transducer [29]. With the subject in the CCFT position, the air-filled pressure sensor unit was set to the pressure of measured maximum strength, and ultrasound imaging of the longus colli was performed in each of the four steps (0°, 20°, 40°, and 60°) of the pre-installed guideline of gaze direction. The SCM was measured [30]. At that time, the movement of the neck joint was limited to allow only a change in gaze direction. In each step, a 1-min rest period was provided to prevent the subject from experiencing muscle fatigue during the muscle contraction process. The transducer was positioned at the C5–C6 level, 1.5 cm below the occipital process [45]. To calculate the thickness of the longus colli and SCM, three lines were drawn on the muscle at the farthest distance from the inside of the fascia boundary, and the average value was taken (Figure 2). The measured thickness was normalized as follows, according to a previous research method [46,47,48]: DNF contracted ratio 20° = DNF20°-DNF0°; DNF contracted ratio 40° = DNF40°-DNF0°; DNF contracted ratio 60° = DNF60°-DNF0°; SCM contracted ratio 20° = SCM 20°-SCM 0°; SCM contracted ratio 40° = SCM 40°-SCM 0°; SCM contracted ratio 60° = SCM 60°-SCM 0°; and DNF to SCM ratio = DNF-SCM.

### 2.3. Data Analysis

The data collected in this study was analyzed using SPSS version 20.0 (IBM, New York, USA). One-way repeated measures ANOVA was used to analyze muscle activity according to the gaze direction angle of the DNF and SCM. For post-hoc comparison of differences between values, tests of least significant difference (LSD) were used. The statistical significance level was set to 0.05 or less.

## 3. Results

Participant characteristics are presented in Table 1. The DNF-SCM varied significantly according to gaze direction (*p* = 0.04), and post-hoc analysis revealed that the DNF-SCM for gaze directions of 20° was significantly different from that for a gaze direction of 0° (*p* = 0.009) and 40° (*p* = 0.041) (Table 2) (Figure 3). There was no significant difference in DNF activation according to gaze direction, and there was a significant difference in SCM activation according to gaze direction (*p* = 0.001). Post-hoc analysis revealed that SCM 60° was significantly different from SCM 20° (*p* = 0.001) and SCM 40° (*p* = 0.001) (*p* < 0.05) (Table 2).

## 4. Discussions

This study investigated the effect of gaze direction on DNF and SCM activation during the CCFT in CNP patients. CNP patients require a specific low-load CCFT, because specific damage has been confirmed for DNF muscle activation [22]. The CCFT promotes DNF activation, inhibits SCM activation, and is effective in restoring function in CNP patients [21]. The CCFT is generally performed while looking slightly downward [19]. For individuals without CNP, DNF and SCM activation has been shown to change according to gaze direction in the CCFT [30]. In this study, changes in LC and SCM muscle thickness were measured using ultrasound images. Ultrasound images have the advantage of being convenient and safe to use and being able to check muscle changes in real time. In previous ultrasound imaging studies, it was demonstrated that DNF and SCM had a negative correlation for CNP and forward neck posture. Moreover, it has proven a correlation between DNF and gaze during CCFT in asymptomatic people. Therefore, in this study, we tried to prove the correlation between gaze direction and DNF using ultrasound images.

Our study confirms that DNF and SCM activation also changes according to gaze direction in CNP patients during the CCFT. Head and eye movement transmits posture and vision information to the nucleus reticularis gigantocellularis (NRG) in the subcortical region, which activates cervical muscles through the motor neurons of the cervical spinal cord, leading to postural stability [49]. In particular, the DNF and proprioceptors of the eye movement muscles maintain the head position and adjust postural balance in real time. Eye tracking, in which gaze direction follows the direction of head movement, activates the central mesencephalic reticular formation within the subcortical region, and information is derived from the part of the brain stem that controls eye movement—the occulari system—and the NRG’s head movement region. This information is delivered to the center to control eye–head movement [50]. Therefore, it is judged that the activity of the longus colli and the SCM should also increase as gaze direction changes during the CCFT, as shown in this study. 

In this study, CNP patients had a DNF to SCM ratio of 2.15 at a gaze direction below 20°, which was significantly higher than at a gaze direction below 40° (1.98) and at a gaze direction below 60° (1.74). These results mean that DNF activation is most efficient when the gaze direction is below 20°.

The DNF contracted ratio maintained a similar value as the degree of gaze direction increased, but, in contrast, the SCM contracted ratio increased as the degree of gaze direction increased. In a study that applied the CCFT to individuals without CNP, the SCM contracted ratio increased as the degree of gaze direction gradually increased to below 20° and below 40° [30]. In previous studies, SCM activation showed a specific pattern of increase with increasing gaze direction and direction of rotation angle (0°, 15°, 30°, and 45°) [31]. In the previous study, when the central part of the computer monitor was within a gaze direction of below 40°, the activity of the SCM, trapezius muscle, etc., and the head tilt angles were increased more than when it was within a gaze direction of below 15° [51]. In addition, in the case of the CCFT, SCM activation showed the lowest activity at below 45° in a study comparing gaze directions above 45°, above 90°, and below 45° [36]. In the case of the CCFT, in most studies, SCM activation is affected not only by neck motion, but also by gaze direction, and in particular, there is a positive correlation with the degree of gaze direction.

On the other hand, in the present study, DNF activation did not correlate significantly with gaze direction, and the gaze direction was higher than that of 20° and 60° to 40°. In the previously reported study, in individuals without CNP, DNF activation during the CCFT increased as the degree of gaze direction increased to 20° and 40° [30]. In the case of gaze direction to the jaw, longus colli activation increased by 10% [52]. In contrast, in previous studies, longus colli activation during the CCFT was not significantly correlated with gaze direction (above 45°, above 90°, and below 45°) [36]. In addition, in other studies, the deep multifidus muscle did not show a significant pattern in terms of gaze direction or direction of rotation (0°, 15°, 30°, and 45°), similar to this study [31]. As such, in most studies, it can be seen that DNF activation increases when the gaze direction is directed downwards. However, it is understood that no specific pattern appears. 

Based on these results, a specific pattern was found for gaze direction and SCM activation during the CCFT, but a specific pattern was not found for DNF activation, although an increase in amount was observed. Therefore, in the case of the CCFT, it is necessary to focus on SCM activation control, and recognize that a gaze direction below 40° causes excessive SCM activation, reducing the efficiency of DNF-SCM activation. Therefore, in order to improve DNF activation in the CCFT, we recommend that the gaze direction be about 20°. This study has limitation. First, the tests were conducted only in patients with neck pain. A comparative study with asymptomatic individuals and severe CNP patients is needed in the future, by selecting subjects without severe symptoms of CNP. Second, we had difficulty to completely limit the head movement. The accuracy of gaze direction was improved by using a pre-installed device, but it was still difficult to control fine movements, because the movement of the head was not fixed.

## 5. Conclusions

This study investigated the effect of gaze direction on DNF and SCM activation during the CCFT. We found that as gaze direction gradually increased, SCM activation increased significantly, and the DNF to SCM ratio gradually decreased. Therefore, in order to increase the efficiency of DNF activation during the CCFT, it is necessary to control SCM activation, and a gaze direction of below 20° is the most efficient. It is expected that the results of this study will be relevant for DNF training of CNP patients in a clinical environment. 

## Figures and Tables

**Figure 1 healthcare-08-00449-f001:**
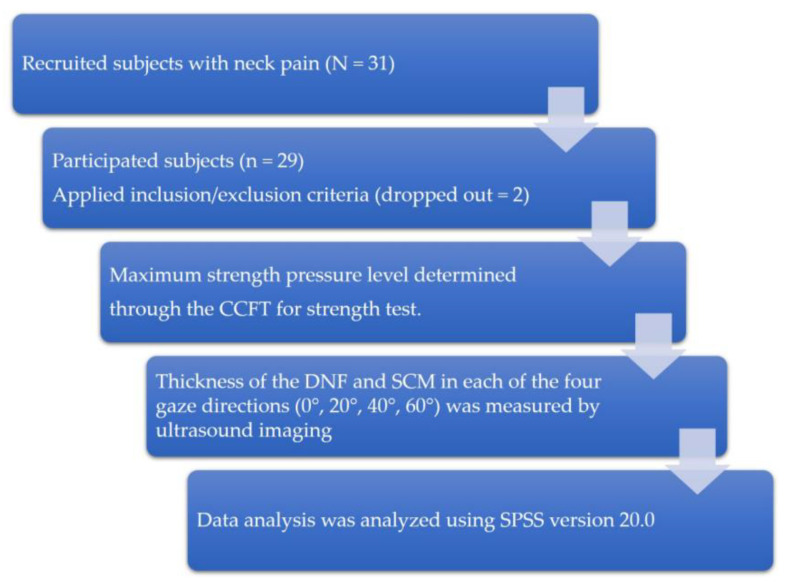
Study protocols.

**Figure 2 healthcare-08-00449-f002:**
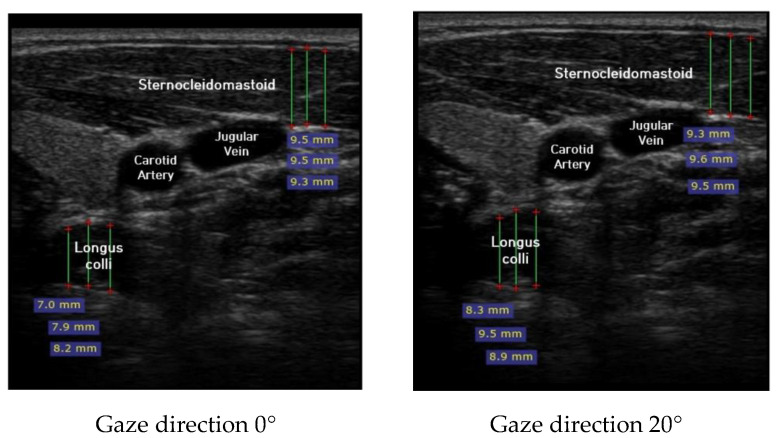
Ultrasound imaging for gaze directions.

**Figure 3 healthcare-08-00449-f003:**
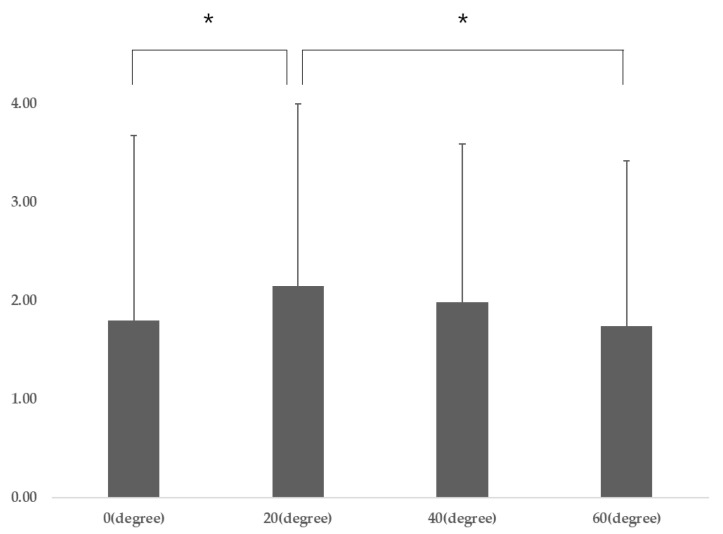
DNF-SCM (different ratio) for gaze directions (post-hoc results). Abbreviations: SCM, sternocleidomastoid; DNF, deep neck flexors; F, F-value * *p* < 0.05 and ** *p* < 0.01 vs. DNF-SCM for gaze directions of 20°.

**Table 1 healthcare-08-00449-t001:** Participant characteristics.

Variables	General Characteristics
Gender	Male (10)/Female (19)
Age (year)	26.59 ± 2.5
Height (cm)	165.53 ± 6.31
Body weight (kg)	59.79 ± 9.5
Maximal strength (mmHg)	22.14 ± 2.56
NDI (point)	10.96 ± 3.67
Endurance (seconds)	14.54 ± 5.98

**Table 2 healthcare-08-00449-t002:** The change according to gaze directions.

	0°	20°	40°	60°	F	*p*	η^2^
DNF (cm)	8.58 ± 1.67 ^a^	8.74 ± 1.72	8.72 ± 1.53	8.76 ± 1.53	0.853	0.47	0.03
SCM (cm)	6.78 ± 1.58	6.60 ± 1.64	6.74 ± 1.57	7.02 ± 1.60	9.428	0.00	0.252
DNF-SCM (different ratio)	1.80 ± 1.88 **	2.15 ± 1.85	1.98 ± 1.61	1.74 ± 1.68 *	2.897	0.04	0.094
DNF contracted ratio		0.16 ± 0.57	0.14 ± 0.61	0.17 ± 0.82	0.037	0.96	0.001
SCM contracted ratio		−0.19 ± 0.43 ^††^	−0.04 ± 0.45 ^††^	0.24 ± 0.55	18.054	0.00	0.392

Values of DNF, SCM, different ratio, and contracted ratio are means ± standard deviation. Abbreviations: SCM, sternocleidomastoid; DNF, deep neck flexors; η^2^, partial eta squared; F, F-value * *p* < 0.05 and ** *p* < 0.01 vs. DNF-SCM for gaze directions of 20°; F, F-value ^†^
*p* < 0.05 and ^††^
*p* < 0.01 vs. SCM contracted ratio 60°.

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
