# Peer review of "The Importance of Optimal Gaze Direction on Deep Neck Flexor Activation in Chronic Neck Pain"

_healthcare, 2020, doi:10.3390/healthcare8040449_

Round 1
Reviewer 1 Report
I believe necessary references of the reliability and validity of the tests used to collect the variables of strength and resistance.
In the material and methods section, an image of the protocols to collect the variables would help the reader. The type of study carried out is not indicated anywhere within the material and methods paragraph.
No limitations were found in the study?
Author Response
RESPONSE TO THE ONE REVIEWER COMMENTS
Reviewer(s)' Comments to Author:
Reviewer: 1
I believe necessary references of the reliability and validity of the tests used to collect the variables of strength and resistance.
Response: Thank you for your suggestions. As you recommended, we added references and explanation about the reliability and validity. Please see the line 107, 108, 127, 358 in the revised manuscript.
In the material and methods section, an image of the protocols to collect the variables would help the reader. The type of study carried out is not indicated anywhere within the material and methods paragraph.
Response: We highly appreciate the detailed valuable comment. We added explanation about the type of study. Please see the line 60 in the revised manuscript. And We added image of the protocols. Please see the line 77 or “figure 1” in the revised manuscript.
No limitations were found in the study?
Response: We highly appreciate the detailed valuable comment. We added explanation about limitation. Please see the line 230, 233 in the revised manuscript

Reviewer 2 Report
The paper aims to identify the optimal gaze direction in the cranio-cervical flexion test (CCFT) for enhancing the deep neck flexors (DNF) in chronic neck pain (CNP) patients, so that CNP patients can recover normal neck function.
In the first place, regarding issues of form, we must say that it is an article that meets the criteria established by the journal, its format, presentation, structure and organization of information being adequate, distributing the information in each of the sections in a balanced and coherent way, from the introduction to the conclusions.
Secondly, regarding content aspects, this research is pertinent, since it responds to the current needs that society demands. There are many research studies that abound about chronic neck pain. But previously it has not been studied the optimal gaze direction in the CCFT for enhancing the DNF in CNP patients. In this case, this objective is achieved with great efficiency, since the topic discussed bets on the resolution of a problem that is very important at the present time. This is due to the fact that this research problem defines an epidemiological, clinical, therapeutic reality and a health problem, with great socio-health implications.
In this paper it is very clear what problem we are facing and how it can be justified to carry out a paper of these characteristics. On the other hand, the identification of aims is concise, being able to observe how the conclusions of the study are clearly linked to the objectives of this work. The manuscript correctly reflects the information handled.
The authors are only recommended to add the calculation of the effect size (partial eta squared ηp2), of the observed differences, in the results section.
Author Response
Review 2.
The paper aims to identify the optimal gaze direction in the cranio-cervical flexion test (CCFT) for enhancing the deep neck flexors (DNF) in chronic neck pain (CNP) patients, so that CNP patients can recover normal neck function.
In the first place, regarding issues of form, we must say that it is an article that meets the criteria established by the journal, its format, presentation, structure and organization of information being adequate, distributing the information in each of the sections in a balanced and coherent way, from the introduction to the conclusions.
Response: We highly appreciate the detailed valuable comment. We have written according to the journal standards.
Secondly, regarding content aspects, this research is pertinent, since it responds to the current needs that society demands. There are many research studies that abound about chronic neck pain. But previously it has not been studied the optimal gaze direction in the CCFT for enhancing the DNF in CNP patients. In this case, this objective is achieved with great efficiency, since the topic discussed bets on the resolution of a problem that is very important at the present time. This is due to the fact that this research problem defines an epidemiological, clinical, therapeutic reality and a health problem, with great socio-health implications.
Response: We highly appreciate the detailed valuable comment.
In this paper it is very clear what problem we are facing and how it can be justified to carry out a paper of these characteristics. On the other hand, the identification of aims is concise, being able to observe how the conclusions of the study are clearly linked to the objectives of this work. The manuscript correctly reflects the information handled.
Response: We highly appreciate the detailed valuable comment.
The authors are only recommended to add the calculation of the effect size (partial eta squared ηp2), of the observed differences, in the results section.
Response: We highly appreciate the detailed valuable comment. We added the calculation of the effect size (partial eta squared ηp2). Please see the line 174 or Table 2 in the revised manuscript.

Reviewer 3 Report
Thank you for submitting a good study to restore neck function in patients with neck pain. The subject of this study is the cranio-cervical flexion test (CCFT) with gaze direction. Chronic neck pain patients have weak deep neck flexors and a hyperactive sternocleidomastoid. The cranio-cervical flexion test promotes activation of the deep neck flexors and decreases activity of the sternocleidomastoid, promoting pain recovery, but recent research suggests that in patients without chronic neck pain, sternocleidomastoid activation increases with increasing gaze direction. In patients with chronic neck pain, it is important to accurately identify changes in deep neck flexor and sternocleidomastoid activity according to gaze direction so that the appropriate gaze direction can be prescribed. Your study plugs this gap in knowledge, and therefore makes a novel contribution to existing research.
You find that as gaze direction gradually increases, sternocleidomastoid activation significantly increases, and the deep neck flexor to sternocleidomastoid ratio gradually decreases. Such findings allow you to recommend that a gaze direction of below 20° is the most efficient for controlling sternocleidomastoid activation and thus increasing the efficiency of deep neck flexor activation during the cranio-cervical flexion test. Your findings will be of particular importance in a clinical setting, where they can inform the deep neck flexor training of chronic neck pain patients.
Your manuscript addresses an important issue in the study of chronic back pain and has clinical relevance. However, in places there was a lack of detail and occasional lack of clarity.
Line 11~12. Please describe simply the pressure measurement procedure.
Line 60. Please describe your recruitment information (how to recruit).
Line 66. Please delete “와”.
Line 77, 86. Please change the order of Outcome measure as follow.
- 2.1. Joint mobility test, 2.2.2. CCFT for strength.
Line 104, 134 141. Please unify the decimal places for all values ​​in the text and table.
Line 107. Please describe the purpose of the NDI evaluation method.
Line 129. Please edit the picture so that the letters are clearly visible.
Line 160. The discussion should be described by dividing into discussion on methods and results. Your discussion need additional discussions on the method of research.
Best Regards!
Author Response
Review 3
Thank you for submitting a good study to restore neck function in patients with neck pain. The subject of this study is the cranio-cervical flexion test (CCFT) with gaze direction. Chronic neck pain patients have weak deep neck flexors and a hyperactive sternocleidomastoid. The cranio-cervical flexion test promotes activation of the deep neck flexors and decreases activity of the sternocleidomastoid, promoting pain recovery, but recent research suggests that in patients without chronic neck pain, sternocleidomastoid activation increases with increasing gaze direction. In patients with chronic neck pain, it is important to accurately identify changes in deep neck flexor and sternocleidomastoid activity according to gaze direction so that the appropriate gaze direction can be prescribed. Your study plugs this gap in knowledge, and therefore makes a novel contribution to existing research.
Response: We highly appreciate the detailed valuable comment.
You find that as gaze direction gradually increases, sternocleidomastoid activation significantly increases, and the deep neck flexor to sternocleidomastoid ratio gradually decreases. Such findings allow you to recommend that a gaze direction of below 20° is the most efficient for controlling sternocleidomastoid activation and thus increasing the efficiency of deep neck flexor activation during the cranio-cervical flexion test. Your findings will be of particular importance in a clinical setting, where they can inform the deep neck flexor training of chronic neck pain patients.
Response: We highly appreciate the detailed valuable comment.
Your manuscript addresses an important issue in the study of chronic back pain and has clinical relevance. However, in places there was a lack of detail and occasional lack of clarity.
Line 11~12. Please describe simply the pressure measurement procedure.
Response: We highly appreciate the detailed valuable comment. We added explanation about pressure measurement procedure. Please see the line 11~12 in the revised manuscript.
Line 60. Please describe your recruitment information (how to recruit).
Response: We highly appreciate the detailed valuable comment. We added explanation about how to recruit. Please see the line 72 in the revised manuscript.
Line 66. Please delete “와”.
Response: We highly appreciate the detailed valuable comment. We deleted “와”. Please see the line 67 in the revised manuscript.
Line 77, 86. Please change the order of Outcome measure as follow.
- 2.1. Joint mobility test, 2.2.2. CCFT for strength.
Response: We highly appreciate the detailed valuable comment. We changed it. Please see the line 80, 98 in the revised manuscript.
Line 104, 134 141. Please unify the decimal places for all values ​​in the text and table.
Response: We highly appreciate the detailed valuable comment. We changed it. Please see the line 174 or table 2.
Line 107. Please describe the purpose of the NDI evaluation method.
Response: We highly appreciate the detailed valuable comment. We added explanation about the purpose. Please see the line 130 in the revised manuscript.
Line 129. Please edit the picture so that the letters are clearly visible.
Response: We highly appreciate the detailed valuable comment. We changed the picture. Please see the figure 2 in the revised manuscript.
Line 160. The discussion should be described by dividing into discussion on methods and results. Your discussion need additional discussions on the method of research.
Response: We highly appreciate the detailed valuable comment. We added explanation about discussions on the method of research. Please see the line 182~ 188 in the revised manuscript.
